# Digital Forms of Commensality in the 21st Century: A Scoping Review

**DOI:** 10.3390/ijerph192416734

**Published:** 2022-12-13

**Authors:** Maína Ribeiro Pereira-Castro, Adriano Gomes Pinto, Tamila Raposo Caixeta, Renata Alves Monteiro, Ximena Pamela Díaz Bermúdez, Ana Valéria Machado Mendonça

**Affiliations:** 1Graduate Program in Collective Health, University of Brasília, Brasília 70910-900, Brazil; 2Study Group in Communication, Culture, Technology, and Food—COMTELA, School of Education and Health Sciences, University Center of Brasília, Brasília 70790-075, Brazil; 3Residency Program in Public Health Policy Management, Fiocruz School of Government, Fiocruz, Brasília 70904-130, Brazil; 4Department of Nutrition, School of Health Sciences, University of Brasília, Brasília 70910-900, Brazil

**Keywords:** eating together, food studies, digital commensality, food practices, technology, social dining

## Abstract

The social act of eating together has been influenced and mediated by technologies in recent decades. This phenomenon has been investigated in different academic fields, but the topic is still in an incipient dimension, and there is a lack of consensus regarding terminology and definitions. The study aimed to characterize the main scientific findings regarding digital forms of commensality in the 21st century and to identify possible relationships between these practices and public health. A scoping review was conducted to identify papers published in different languages between 2001 and 2021. A total of 104 publications that combined commensality and technology in all contexts were included. Most studies were qualitative; from the Design and Technology field; used social media and video platforms or prototypes/augmented reality gadgets; and used different terms to refer to digital forms of commensality, allowing the analysis of the construction of field definitions over time. The intersections with health were observed from impacts on family/community engagement, culinary skills development, and mental health and eating habits. These practices also structured specific social interactions, such as virtual food communities and commensality, during the COVID-19 pandemic. This paper indicates the consistent growth of these practices and recommends the development of future research for theoretically and longitudinally deeper evaluations of the impacts of these new ways of eating together, especially regarding their effects on human health.

## 1. Introduction

The act of eating is a human daily practice that is not limited to satisfying vital physiological needs, but also implies a dimension that is socially constructed, shared, and crossed by social markers [1,2,3]. Sharing meals is part of humanity’s civilizing process and represents food cultures and social settings around the world [4]. This phenomenon is known as commensality, a term that is defined as the act of eating together and has its etymological origin in the Latin word “Mensa”, which means to live together at the table. Therefore, commensality is considered as the act of sharing a meal with others [5,6].

In the literature, studies on commensality are historically concentrated mainly in the field of Human and Social Sciences [6] and propose theorization of this concept and analyze its different manifestations between cultures throughout human evolution. The scientific investigations carried out in this field aimed, among other things, to describe and analyze the ways of eating together in everyday and festive situations [7], such as in the family context and in other social relationships; in common spaces, such as at the table, and also outside of common spaces, such as, e.g., eating while sitting on the couch or in bed [8]; and to understand the hierarchy established between the human relationships expressed through this practice (power relations), the hospitality involved, the codes of rules in the regulation of behavior at the table, and the health impacts of eating together with a certain frequency [4,5,6].

In the contemporary era, new debates arise around this phenomenon, especially those considering that lifestyles are constantly changing, bringing new ways of eating, such as a greater frequency of eating outside of the home, the individualization of this practice, and even the questioning of a possible decline in commensality in its traditional format [5]. Facing a society whose virtualization affects even the modalities of being together [9], other forms of commensality begin to be perceived and are greatly influenced by the presence of technology in the daily meals of 21st century societies. Food is now shared in the digital space, and there is an increase in the use of electronic devices to share meals remotely [10]. Examples include the exaltation of the aesthetic values of food, also known as the practice of food porn; the creation of virtual communities whose central theme is food; and online meal broadcasts, such as mukbangs, which gained popularity due to its videos of people eating excessive amounts of food [11,12,13].

In this context, there are reflections [14] about the role that computerized technologies play in society, as well as the understanding of the way societies use these technologies to create cybercultures. Following this type of analysis, contemporary commensalities characterized by technological mediation are interesting for creating narratives and discourses that refer to values, worldviews, and new constructions about the meaning of food and how people or social groups fabricate new relationships mediated by technological devices around a specific context in which such commensalities are configured.

Likewise, for the traditional forms of commensality, these types of digital practices have been described according to their potential to have positive or negative impacts on the social, cultural, and psychological dimensions of human life [12]. They can create solutions for family and community members that are physically distant [15]; they can create a sense of company and reduce feelings of loneliness among people who have individualized routines [16,17,18,19]; and they can also stimulate the excessive consumption of food or interfere with food choices [20].

Therefore, this research had as its main question: What are the main characteristics of the studies on forms of commensality in the 21st century that are related to technologies? It is true that commensality as a research topic is expanding in all of its dimensions, such as in how it is presented and discussed by Jönsson et al. [21], who reveals the diversity of areas of knowledge interested in the theme and proposes concepts related to eating together. Additionally, considering that technological evolution is in constant movement and that it consequently influences the ways of eating reveals the importance of characterizing studies that present this thematic approach and its theoretical contribution to the field of food studies. On the other hand, this theme still has an incipient scientific study that presents concepts that still require better characterization and understanding. For these reasons, a scoping review was conducted in order to characterize the main findings in the scientific literature regarding digital forms of commensality in the 21st century as well as to identify emergent topics related to these practices with public health.

This scoping review was guided by the strategy of defining the Participants, Concept, and Context—PCC [22]. As for the participants, studies on commensality for all populations were included, with no restrictions. Regarding the concept, publications that focused on the act of eating together associated with technology were considered. Sharing the act of eating was considered both for face-to-face forms in which technology is used at some point during the meal (when alone or with others) and for virtual ways of sharing food (synchronously or asynchronously). The contexts were scientific studies that addressed the topic in the 21st century in all possible circumstances.

In this study, commensality was considered as the moment when interaction between people occurs through the sharing of food and/or the act of eating together, enabling the exchange of ideas, opinions, stories and, therefore, fostering social relationships [6]. With regard to technologies, communication and information technologies were considered, in the context of cyberculture, whose ubiquity element allows the intermittent circulation of information made by everyone and for everyone through the use of different electronic artifacts [14].

## 2. Materials and Methods

### 2.1. The Scoping Review Approach

In order to characterize commensality in the contemporary context, we identified relevant existing studies to propose ways to deepen the scientific investigation of this phenomenon. Scoping reviews identify the nature and extent of the available scientific literature, identifying possible gaps in research, and analyzing the need for a systematic review [23]. It is also important to differentiate a scoping review from other classic literature reviews because it does not include an analytical interpretation of the texts. Scoping reviews map investigations are still being developed and try to identify concepts and terms that are being used by different authors and studies [24]. Therefore, when considering new forms of commensality, sometimes called “digital commensality,” “virtual commensality,” or “remote commensality,” this study aimed to better clarify definitions of this form (or forms) of commensality, presenting complex concepts that are currently constructed.

This scoping review was conducted based on the methodological steps proposed by the Arksey and O’Malley model [25], improved by Levac et al. [23]: (1) identification of the research question; (2) identification of relevant studies; (3) selection of studies; (4) data tabulation; and (5) collection, summary, and description of results. The sixth step (optional), which refers to consulting specialists on the subject, was not used in this study.

In order to develop this research, a protocol was developed *a priori* based on recommendations from the Joanna Briggs Institute [26]. Additionally, the PRISMA Checklist for Scoping Reviews (Appendix A) was applied to adequately contemplate all of the fundamental items required to report a scoping review [27]. The protocol was not registered online, but it is available on request.

### 2.2. Study Identification and Eligibility Criteria

Only studies published between January 2001 and December 2021 were included. Studies that were still being conducted were also included, with the objective of mapping the full academic scope of the technologically mediated forms of commensality. Studies could be quantitative, qualitative, mixed, or even opinion articles. Therefore, in addition to selecting scientific articles, the study selected dissertations and theses, conference proceedings, reviews, essays, and gray literature. The electronic databases PubMed, EBSCOHost, SCOPUS, Web of Science, PROQUEST, and Google Scholar were used to search for and identify studies, including publications written in Portuguese, English, Spanish, French, and Italian. The decision to use these languages was motivated by the understanding that (traditional) commensality is a concept that is historically and sociologically investigated in French and Italian studies and due to the interest in identifying possible studies (in Portuguese and Spanish) from Latin America since the researchers are Brazilian. The final database search was performed on 28 June 2022. We excluded studies that we were not able to access and that did not match the search theme or the scope of the study.

The keywords used to search for the studies are described in the following syntax: “((“commensality”) AND (“technology” OR “digital communication” OR “internet” OR “mobile” OR “communication technologies”)) OR (“Digital commensality” OR “Virtual commensality” OR “Remote commensality” OR mukbang), and the search terms were also translated into the other languages.

### 2.3. Study Selection, Data Extraction and Analysis

Two independent reviewers (MPC and AP) read the titles, abstracts, and full texts, in order to engage in further discussions regarding whether to include studies. In cases of divergence in study selection and data extraction, a third reviewer was called (TC). The data were extracted from the selected studies by filling out an instrument in Google Sheets developed by the researchers (MPC, AM, and XB). MPC and AP recorded specific information on each study, such as the authors, year of publication, country of origin, type of study, objectives, commensality terms used, forms of eating together represented, type of technology used in the practice of commensality, and health-related aspects.

The description of the results related to the research objective, and the question is presented below. Both descriptive and thematic analyses were performed to present the results found. In this sense, we pointed out the frequency of the types of publications and studies; the target audience; language; country of origin; year of publication; disciplinary areas; and types of technologies studied. To conceptualize the digital forms of commensality, we listed all of the terms used in the publications to characterize it, and afterwards, we categorized it in the groups presented in a table. We also made a timeline of the occurrences of these terms to show the semantic impact on the thematic construction of contemporary forms of commensality.

Additionally, a narrative synthesis was performed to describe the qualitative analysis. It was possible to identify, through inductive analysis, different categories to represent the health topics mentioned in the context of the practice of digital forms of commensality as well as to characterize the ways of eating together represented in the studies.

## 3. Results

### 3.1. Search Results and Study Characteristics

A total of 17,306 publications were found in the databases. After removing 5522 duplicates, 11,784 titles were screened, 501 abstracts were screened, and 206 documents were fully evaluated. Of these, 104 publications were included in the review for result extraction and analysis based on the eligibility criteria. The flowchart below systematizes the search strategy and the study selection process (Figure 1). Appendix A presents a table with the 104 publications analyzed, including the main information collected and systematized for the study.

This review included studies published in scientific journals (*n* = 64, 61.5%), annals of events (*n* = 24, 23.1%), master’s theses (*n* = 10, 9.6%), doctoral dissertations (*n* = 5, 4.8%), and books (*n* = 1, 1.0%). Of the 104 publications that met the inclusion criteria, 53 were qualitative studies (51.0%), 20 were quantitative studies (19.2%), 14 had mixed research methods (13.5%), and 17 were reviews, opinion texts, critical essays, and experience reports (16.3%). Different data collection and analysis methods and techniques were used, such as digital ethnography, survey, interviews, focus groups, participant observation, topic analysis, semiotics, experiments, and prototype co-creation. As for the populations analyzed, we included studies that analyzed the behaviors of young people (*n* = 12, 11.5%), families (n= 10, 9.6%), content producers (*n* = 3, 2.9%), older people (*n* = 2, 1.9%), immigrants (*n* = 2, 1.9%), and the market (*n* = 2, 1.9%).

The selected studies were written in English (88.5%, *n* = 92), Spanish (3.8%, *n* = 4), Italian (2.9%, *n* = 3), Portuguese (2.9%, *n* = 3), and French (1.9%, *n* = 2). Publications were from institutions located on 5 continents in 29 countries. The USA (*n* = 19, 14.3%), the UK (*n* = 18, 13.5%), and Australia (*n* = 15, 11.3%) are the three countries with the largest number of studies on the subject. Figure 2 presents the distribution of publications identified in each country.

Although we analyzed articles on the subject published since the beginning of the 21st century, only studies from the last 10 years were found (Figure 3). Figure 4 presents the areas of knowledge investigating digital forms of commensality. The areas were identified based on the department at the institutions to which the author was linked. Most studies were in the fields of Design and Technology, Human and Social Sciences, and Communication. Studies whose authors were linked to different areas were classified as interdisciplinary. However, in most cases, the authors of the same publication were from the same area, even from different institutions. Figure 3. Number of studies published by year (*n* = 104).

### 3.2. Understanding Commensality

The main terms used to refer to the form of commensality related to the use of technology were mapped in each of the selected studies; the results were grouped into categories. However, the absence of a specific concept to characterize the context of technology-mediated commensality was observed in many studies. Thus, terms related to commensality were often used in general (*n* = 37, 35.6%), without differentiation between traditional or technology-mediated practices.

In the analyzed scope, few studies tried to conceptualize these practices. Spence et al. [12] proposed the term Digital Commensality to refer to a set of practices that combined food and the use of digital technologies, such as the mediation of meals by videoconference, the use of interactive digital agents—such as robots—and the practice of eating alone while using the Internet or watching someone else eat—the mukbang. Concurrently, Grevet et al. [16] used the term Remote Commensality to refer to the use of companionship systems to gather geographically distant diners. Medina and Bayre [28], in turn, concluded that the use of social networks to share images can be understood as an active part of the eating experience, given the social and symbolic value contained in the practice; they called this phenomenon Virtual Commensality. Niewiadomski et al. conceptualized the term Computational Commensality as “technology which computationally addresses various social aspects of food and eating” [29] (p. 1).

In 12 studies, terms were identified for the act of eating together remotely (11.5%), which was associated with the virtual sharing of meals by videoconference, with each diner located in a different physical environment. Digital commensality was cited in 14 studies (13.5%). This term considers the practice of eating together in an asynchronous format, whether mediated by text, image, or video posts or instant messages. Mukbangs and cookbangs (*n* = 25, 24.0%) are ways of sharing eating and cooking activities in live broadcasts with an audience. Although the selected publications are mostly in the field of Design and Technology, only 10 studies cited terms of commensality centered on interactive devices, such as eating with a robot or computational commensality (9.6%).

Table 1 shows the organization of the terms used in the sample publications based on the conceptualization of four forms of technology-mediated commensality. The concepts were coined partially from the appreciation of previous attempts to characterize these commensality practices and from the effort to synthesize ideas and terminological approaches in the sample according to their similarities and differences.

At the same time, observing the creation and recurrence of the use of terms related to forms of contemporary commensality over the years, the progressive ways in which authors started to coin terms that cataloged historically unprecedented commensal practices, and also to reproduce them, stands out, sometimes from direct references to previous publications and sometimes signifying them. In Figure 5, the occurrences related to the use of contemporary commensality terms during the time frame of the research are organized chronologically. It can be noted that the terms that indicate traditional forms of commensality were not included in the figure, in view of their small semantic impact on the thematic construction of digital forms of commensality. Additionally, it is noteworthy that, between 2011 and 2013, the terms used to refer to these practices were more general and that, in the following years, the academic discussion had moved towards a scenario of increasing specification and the identification of different social processes and practices between diners and digital technologies.

### 3.3. The Technological Landscape of Commensality

All of the analyzed texts directly used or theoretically explored at least one form of digital technology, and their presentations and delimitations varied according to the objective of each study. The ones dedicated to the development of technological prototypes that could be used to mediate the practices of commensality often gave more details regarding the conceptual and technical aspects of their technologies, while studies that focused on understanding the behavior of users of popular digital platforms or gadgets associated with commensality tended to have simpler conceptualizations and explanations of these technologies.

Studies on social networks, applications, blogs, and websites contributed to 25.6% (*n* = 32) of the occurrence of technologies in the total sample. In these cases, the electronic devices used to access digital platforms assumed a secondary role or were not mentioned in the publications.

Similarly, studies on the behavior of hosts and audiences in videos and live streams on video platforms, such as YouTube and AfreecaTV, represented 18.4% of the sample (*n* = 23) of use of technology related to food, with a material scope guided mainly by the investigation of mukbang and other forms of content production with virtual or digital commensality.

Additionally, 15.2% of studies (*n* = 19) tried to build or validate disruptive prototypes and to present new ways of mediating social experiences during eating with the use of virtual reality (VR) headsets [30], tactile sensation stimulators [65], and interactive tableware [26]. In a similar proportion, 13.6% of the studies (*n* = 17) used mobile devices, especially smartphones and tablets, which could be explored according to their individual, collective, and interactional uses. Videoconferencing tools were used in 11.2% of the analyzed sample (*n* = 14), with commercial and popular products such as Zoom or with more experimental forms of technology.

Concurrently, television sets were addressed in 4.8% (*n* = 6) of the sample. However, more than half of these publications addressed the use of television sets with mobile technologies. Robots were also used in 4.8% of the sample (*n* = 6), and commercial platforms were used in 4.0% of the sample (*n* = 5), specifically in studies on prototyping or on the investigation of the consumption of robots developed from the virtual environment of gastronomic experience sales. Finally, the use of computers was verified in 2.4% (*n* = 3) of the studies on the domestic lives of families. Table 2 shows the type of technology cited in each of the publications and the absolute number of occurrences throughout the sample.

### 3.4. Commensality and Health

Studies that pointed to issues related to public health in the context of the practice of digital forms of commensality were identified. These were categorized as shown in Table 3.

Studies on the use of technologies during family meals to promote interactions and to connect distant family members or as a space for community interactions around food were classified as “family and community engagement.” Studies on the use of digital technologies as a way of acquiring and developing culinary skills were classified as the “development of culinary skills.” Studies associating the use of social networks to practice the local food culture were classified as “appreciation of the local food culture.” Studies on digital media as ways of disseminating sustainable or unsustainable discourses and practices—such as increased food waste motivated by mukbang consumption—were categorized as “(un)sustainability practices.” The category “formulation of public health policies” included studies that characterized the use of digital technologies as a target for public health and food policies or that addressed the issue of the regulation of such practices with the objective of protecting the population from practices that could harm health and healthy and adequate eating. Finally, categories were created to group studies on the impact of the use of digital technologies in the food context on mental health and on the eating habits of populations and individuals, in which, respectively, the results reported the advantages of the intersections between the digital world and food, such as reducing loneliness and improving eating habits, and the disadvantages, such as encouraging excessive eating and maintaining behaviors associated with eating disorders.

### 3.5. Representations of Eating Together

Considering the different technologies used in the studies, it was possible to identify, through inductive analysis, six ways of eating in which social relationships were represented in the studies: (1) eating together remotely and solo-eating with technology; (2) family meals with children using an electronic device at the table; (3) virtual communities about food; (4) sharing and gaming; (5) commercial relationships in digital forms of commensality; and (6) eating together in the context of the COVID-19 pandemic.

These six representations of eating together demonstrate issues that stand out in the studies selected in this scoping review that may reveal trends in digital forms of commensality practices. Eating together remotely is related to the practice of sharing meals by videoconference between people who are physically distant, while eating alone while using electronic devices during the meal refers to exchanging text messages or browsing social media while eating. Different from the context of being alone, many studies have investigated family meals and their use of electronics influencing face-to-face social relationships during the act of eating. Others sought out the communities that formed and interacted only in the virtual environment to share about food. Food content shared on social media and digital games that propose commensality are also identified representations of eating together. In addition, it was noticed that the practice of digital forms of commensality is often linked to a marketing perspective. Finally, some of the studies that highlighted the context of the COVID-19 pandemic were mentioned as enablers of eating together in a digital form.

In that sense, the next results described and organized the representations of eating together identified among the studies.

#### 3.5.1. Eating Together Remotely and Solo-Eating with Technology

Of the digital forms of commensality, eating together remotely has been a strategy for families, couples, and friends to meet and share this moment when they are geographically distant. In this context, food is considered central to maintaining connection with each other, knowing about each other’s routines, and remembering affective experiences from the past [32]. Depending on the technology used, the sense of physical presence and connectivity can be increased to experience this virtual encounter [66]. Additionally, researchers have indicated that having food ready and served at the same time [76] and eating the same kind of meal encourages more dialogue during the remotely shared meal [67].

In situations where the act of eating is performed alone, especially in public places, it is possible to identify the use of technology as a way to minimize the feeling of loneliness and non-belonging [16,17,18,19]. As an example, a study with freshmen students at a university showed they did not like to eat alone in the dormitory cafeterias, and when they did, many used cell phones to exchange text messages to feel better [18]. Alternatively, the practice of eating alone is sometimes desired by the foodie universe, as reported in [17], which showed that the appreciation and criticism of culinary preparations in restaurants is usually an individual event. This does not imply that foodies will not interact with other people, such as restaurant employees and/or the other diners present there, and they also use technology before, during, and after foodie experiences as a way of practicing commensality with their peers. For people who live alone, such as many older people, prototypes of technological systems are being proposed to enable socialization with others during meals, as demonstrated by Korsgaard et al. [30]. In situations with no interactions with other humans, the company of technology was a distraction as well as a form of entertainment while eating alone [77] that influences the time spent eating, the amount of food ingested, and the food choice itself [77,78].

#### 3.5.2. Family Meals with Children Using an Electronic Device at the Table

Family meals, especially with children, were investigated in terms of their face-to-face practices in which at least one member used electronic devices while eating at the table. According to Moser et al. [120], the more people use their cell phones during meals, the more they think it is appropriate for others to do the same. However, their research showed that using cell phones during a meal with children is considered inappropriate. The study by Nelson [33] identified that the use of electronic devices such as cell phones during family meals is negatively related to the feeling of closeness between parents and children. Davis et al. [34] observed family meals at a restaurant to find that, as soon as the family sat at the table, the children started using technology that was interrupted only when the main dishes arrived. In a few cases, the children requested more time with the device. They also noticed that children using earphones interacted less with the family and that competitive use of games and device sharing was associated with discussions at the table between family members. Alternatively, the use of electronic devices was also seen as a strategy to stimulate dialogue during meals [35,36,37,38,39,79], either through games or other conversation strategies.

#### 3.5.3. Virtual Communities about Food

In a few cases, social relationships around food have been established in cyberspace communities. Several social network communities gather immigrants from the same country and work as a stimulus for the dissemination and sharing of legitimate original recipes and dishes from that culture [68] as well as the exchange of images (and memes) to refer to sensory experiences from the past that reveal the food identity of the country of origin [28].

However, there are several other communities whose members meet virtually to discuss and share questions and experiences around their common food preferences, practices, and styles (example, vegetarianism, food intolerances and allergies, fad diets, and even eating disorders) [40,84].

Mobilization in favor of food activism is another relationship established on the Internet, such as in the study by Pimiento [116], who problematized urban legends created around food. Cyberspace communities are not just made of pleasant aesthetic dishes. Some people build communities around sharing “monster dishes,” culinary mistakes and failures, or junk foods. This kind of community points to the existence of a counter-foodie movement against food porn and that praise to what human eyes may process as being unpalatable for eating [117].

Additionally, several mukbang studies show the construction of interactional relationships between live broadcasting participants and a mukbang streamer [20,93,94,96]. These interactions occur through speeches, written texts, performances, actions (such as donations), gestures, facial expressions, and sound and visual effects around eating together. Somehow, all of them contribute to sharing food between those in front of the cameras and those who follow them.

#### 3.5.4. Sharing and Gaming

Food-related content is not only shared in structured virtual communities, but also to other people through social networks to relativize physical presence and persist in commensality in other media [10]. Sharing food photos and videos was identified in online publications through publications analyzed in this present study, which consider the “food porn” phenomenon common in the contemporary society for the practice of eating together, such as in the studies by Tooming [13]; Mendini et al. [41]; and Lavis [40]. This social interaction is a way to register, in the virtual environment, food as an essential element in daily life and in human history, sharing affections, memories, and desires that can be expressed in posts as food diaries and recipes and even to immortalize special events.

As for the moment of the meal in which these images are shared, Weber et al. [42] reported that it can occur before, during, and after eating. People often take pictures of the preparation of food to share with friends and family who are not physically present, to post on their Instagram accounts, or to send to other diners.

Playing video games while eating can be a form of digital commensality. Young people who participate in game events practice commensality during breaks and after video gaming, as shown by Cronin and McCarthy [43]. This population also showed interaction between peers on video game websites and forums, where they also share about food. In the literature, solutions for better interaction during the act of eating together through games, such as VR games [44], in which players mutually cooperate during the act of eating, or play talking to one’s own food through augmented reality [80], are being proposed. Not only games, but other communication facilitators during meals have been proposed, ranging from light resources to express the emotions felt when eating a dish [45] to applications to choose content such as news, photos, or music that will be displayed on screen and commented on during meals [37,39,80].

#### 3.5.5. Commercial Relationships in Digital Forms of Commensality

The opportunity to invest in new businesses inserted in the digital forms of commensality was another aspect represented by eating together—in this case, the relationship between consumers and companies. Shared food sites, called “food Airbnbs,” are a business model in the hospitality industry that facilitates connections between amateur chefs, tourists, and investors to enable diverse gastronomic experiences [10,46]. Another proposal is to invest in the marketing of a particular food product through digital influencers who practice mukbang. The live broadcasting of eating large quantities of food for thousands of spectators is a favorable stage to configure the marketing relationship of eating [97].

Digital forms of commensality were represented in several gastronomic experiences, for example, in studies reporting the practice of cooking together combined with video recordings of these moments [69,70]. However, the organization of social events around food previously prepared through digital platforms [10,83,84,114] has been highlighted. In other cases, events holding online cooking activities and even virtual wine tasting were identified [69]. Eating in artistic performances is also one of the examples found, such as the case of El Somni [49], whose performance consists of a dinner with several technological interactions conducted in various acts and with different dishes. Remote eating also becomes an attraction [121] when included as a scene in a theatrical play with actors in two different rooms that will interact with one another through the connection of the meal held at a distance.

#### 3.5.6. Eating Together in the Context of the COVID-19 Pandemic

Given the context of the COVID-19 pandemic, which began in 2020, some studies have reported forms of commensality practiced through the use of technologies during the lockdown period. Families that participated in the study by Fourat et al. [70] reported virtual meetings with friends to drink together as well as the sharing of photos of food and recipes prepared during social isolation via instant messaging applications. The reason for deciding to eat online during this period was reported by participants from different countries in the study by Ceccaldi et al. [71] as a way to feel closer to friends that they could not physically meet. As intervention strategies, we can mention the creation of a chat group that encourages conversation between older people by approaching the topic of food during this period [87] as well as the organization of digital meal events aimed at diners from different time zones to get them out of their routines and to stimulate the sharing of isolation experiences and discussions about the pandemic in each country through sharing desserts prepared face-to-face [72].

## 4. Discussion

### 4.1. About the Main Study Characteristics

In this scoping review, 104 studies regarding forms of commensality mediated by technologies in the 21st century published between 2011 and 2021 were mapped. The study sought to identify and analyze their main characteristics that corresponded, as well as to conceptualize the themes, the types of technologies used in the studies, the relationships between commensality and health, and finally, the representations of eating together that were portrayed. A high quantity of empirical studies was identified, as were a number of differences between the quantity of studies that used qualitative and quantitative methods, which reveals a window of opportunity for the development of more research that proposes the application of mixed methods and that have longitudinal characteristics to obtain a broader understanding of the phenomenon of contemporary commensalities. In addition, it was possible to identify diversity in the populations studied, from children to the elderly, university students, families, immigrants, factory workers, and internet content producers, which may indicate the capillarity and versatility of possible contexts for deepening new forms of eating together, which, in turn, presented symbolic, affective, and interactional complexity and sophistication, giving rise to concrete and dynamic relationships between participants.

Publications on the subject are mainly concentrated in specific regions of the globe, so there is a need to expand studies and publications from countries and regions that have been less explored so far, such as Africa and Latin America, in order to show how this phenomenon is configured according to cultural, economic, and social conditions that are different from those already found in the studies mapped in this scoping review. Future comparative studies could allow for cultural differences and similarities between countries around digital forms of commensality. Additionally, the geographic aspects of the research analysis scope made it possible to highlight a reasonable number of publications that investigated rural populations or populations outside large urban centers, especially in China and Japan (e.g., the studies [50,87,98]), which sheds light on a phenomenon of internalization in these digital forms of commensality that is concomitant with the process of democratization of the use of digital technologies already mentioned by Vasilenko and Mescheryakova [123] and gives substance to the argument that they have not only developed in highly urbanized niches of society.

### 4.2. Technology as Emerging Scholar Field for Contemporary Commensalities Studies

The research field of Design and Technology, especially in the area of Human–Computer Interaction, stood out in a number of publications on the subject, many of which sought to create and validate various technological prototypes that supported or that were the core of the act of eating. This can be analyzed in the light of McLuhan’s [124] proposition that the tools and channels used to mediate human interactions contribute to the delimitation of their meanings and content, so that the process of creating tools directly crosses the construction and consolidation of practices, rites, behaviors, contexts, and instruments of new ways of eating together. The growing studies in this area of knowledge provide opportunities for the development of new technological products that favor the practice but that are also discussed in an interdisciplinary way by considering their possible effects on society. It should be considered that, in addition to the study being able to characterize different types of technologies used for the practice of commensality, the existing commercial relationship around this phenomenon is also recognized. Considering such context, new products from Big Tech companies, such as the Metaverse from Meta, could offer new research opportunities in the future by stimulating new interactional possibilities and pop culture trends that would characterize the new ways of eating together as a field of study of considerable dynamism and sensitivity to the contemporary pace of life.

### 4.3. A Social Phenomenon with a Conceptual and Symbolic Heterogeneity

Regarding the conceptualization of digital forms of commensality, this scoping review reveals that such practices are constantly growing, in line with technological trends in contemporary society. This reveals a dynamic universe that allows people to share their meals in different ways through the use of technology. However, Table 1 shows that there are differences in how the authors named and conceptualized the forms of commensality in their studies (e.g., the term Digital Commensality was included both in the concept of remote commensality [75] and in the concept of digital commensality [88]). Such situations denote the conceptual and symbolic heterogeneity attributed to contemporary commensality in a configuration that points to the direction of efforts aimed at systematizing such a construct in the field of new forms of commensality. According to the heterogeneous profile of the publications, a series of different terms were found in the repertoire of their respective authors to designate similar or equal practices of commensality associated with the use of technologies. Such phenomena could be explained both by the differences in the technical–scientific language of each of the areas involved as well as by the hypothesis that the digital forms of commensality field still needs to be sharpened in order to create some nomenclature consensus. Nevertheless, the effervescence of the theme in different countries and from different continents and the short period of exploration also seem to operate as important variables that could explain these differences around terms and concepts. Thus, efforts to organize and systematize the studies of this knowledge are central concerns for its maturation.

### 4.4. About Health Topics Identified

Possible relationships between commensal practices and matters of interest to public health were identified in this study, revealing that new studies on the subject should be encouraged so that robust evidence can be used when discussing the process of formulating public policies to promote health. Health and commensality, especially in its traditional form, have been linked in the scientific literature for a long time [125]. On the one hand, it is shown that eating together has a direct relationship with the improvement of health indicators, such as the promotion of healthy eating habits and impacts on mental health [126]. On the other hand, some evidence [127] and food-based dietary guidelines e.g., [128,129] indicate that eating in front of screens should be discouraged because of its negative effects. However, the studies identified in this article indicate and encourage initial discussions of this dichotomy for a theoretical deepening of the associations between health and the other different forms of commensality mediated by technologies.

Based on what has already been studied about health and contemporary commensalities, it is possible to characterize an initial overview of their interactions. The Mukbang, described throughout the sample as the high consumption of ultra-processed foods by its presenters and the regulation of food consumption by its viewers, was sometimes deleterious by increasing the intake of foods high in salt, fat, sugars, and food additives and sometimes had a positive effect by including lonely individuals in communities [94]; in summary, the research showed the potential of a single practice to combine both benefits and harms to health. Nevertheless, considering the differences inherent to the very types of contemporary commensality practices explored in the review, it was observed that publications on mukbang more often indicated health risks, while studies that addressed the use of videoconferencing tools to connect distant diners mostly pointed to the potential of the practice to socially enrich the lives of its participants and to encourage them, with few mentions of negative aspects, such as increased homesickness and feelings of missing family or feelings of guilt for having moved to a location far from their loved ones [86].

Considering that researchers [130] have suggested that digital food environments can influence health, both positively and negatively, implications for policies can be made so that the practice of technology-mediated commensalities become strategies for the application of interventions in the digital environment that focus on health promotion. Some studies in this review have already proposed this, such as the development of games that facilitate eating together (e.g., Arnold; Khot; Mueller [44]), virtual communities for food literacy (e.g., Pimiento [116]), and holding meetings for remote meals with discussions around the topic of food (e.g., Zegarra [72]). However, there is still a lack of evaluative measures that prove the impacts of such recommendation at the political and scientific levels.

### 4.5. Eating Together in a Digital World

Regarding the last section of the results, it was possible to characterize different representations of eating together in an attempt to present an overview of how contemporary commensality practices have been described and analyzed in the academic environment. The various representations of social relations around the act of eating have sometimes evidenced the contemporary lifestyle characterized by individualization [5] and the spectacularization of the subject [131], either through eating alone accompanied by technology or through sharing photos and videos of food and other manifestations associated with a 24/7 culture [132] in which the interaction of eating becomes uninterrupted: it occurs before, during, and after its actual performance.

From this perspective, Han [133] argues that solitude and the digitalization process is in the core of current society constitution and, if we transport his conclusions as a key to explore commensality, it is possible to inquiry important questions about the decline of a traditional commensality presented in past centuries. For him [133], we are in a digital swarm, in which people are connected through digital media, without meaning that they are together, but, ultimately, they are singular. Thus, would the digital forms of commensality just be a mirror of this contemporary social configuration? Among these forms, are there any that really oppose this idea? It is true that eating alone and not together should be an issue to be considered as a field of research. 

Finally, it should be considered that the context of the COVID-19 pandemic was identified in the results as one of the representations of social relations around eating. The health and social context experienced around the world during the pandemic is an evident catalyst for the historical process of digitization of ways of life and, consequently, of ways of eating together and sharing food [73]. Thus, it is expected that, in the near future, more findings will emerge regarding the impact of the COVID-19 pandemic and the concomitant period of social isolation on health, eating habits, and the process of virtualization of behaviors and eating habits of individuals and populations.

### 4.6. Strengths and Limitations

This is the first scoping review that systematizes studies on the different forms of technology-mediated commensalities and is therefore the first study that expands the possibilities of research in this field. The identification of studies was carried out across many databases and in different languages, which may have contributed to the identification of studies from different contexts and origins. The fact that the term commensality is not yet a descriptor indexed in databases was not an impediment to the identification of the main studies related to the research problem. Furthermore, this review made it possible to identify the variety of terms used in this expanding field to conceptualize the various forms of commensality that are currently practiced.

On the other hand, this scoping review has limitations. It was not possible to characterize all of the studies in relation to their applied methodological techniques given that studies published in different media, including the gray literature, were included and did not provide accurate details of their methods and/or were still under development. Regarding the protocol for carrying out this scoping review, it was not possible to register it a priori in order to make it public. Despite the possibility of consulting specialists to include more studies in the field as well as to add new references from the selected studies themselves, this stage of collection was not carried out for this research. However, it is believed that it was possible to present a very representative overview of the existing studies on the subject in question.

## 5. Conclusions

This scoping review characterized the main findings of the scientific literature related to digital forms of commensality in the 21st century and identified emergent topics about these practices and public health. The work presented here highlights the multi- and interdisciplinary potential that commensality has as a research topic and how its forms have been conceptualized in an increasingly specific perspective, therefore presenting different terms to characterize its practices. The variety of types of technologies mapped in the results demonstrate how much this is being inserted into the act of shared eating in an even more digital world. Thus, one must consider that such forms of commensality relate to health through themes such as family and community engagement, culinary skills development, the appreciation of local food culture, sustainability practices, and public health policymaking and their impacts on mental health and eating habits. Furthermore, the social relations around eating together as mediated by technologies could be characterized by the act of eating together remotely and eating alone with technology; family meals during which electronic devices are used at the table; virtual food communities in which both food and gaming are shared, and the commercial relationship established in these ways; and even eating together in the context of the COVID-19 pandemic. In conclusion, this review presents the existing literature on the subject in a systematized manner and highlights the need for future studies that deepen reflections on the role of socialization provided by digital forms of commensality and that contribute to our understanding of the impact of these practices on human health.

## Figures and Tables

**Figure 1 ijerph-19-16734-f001:**
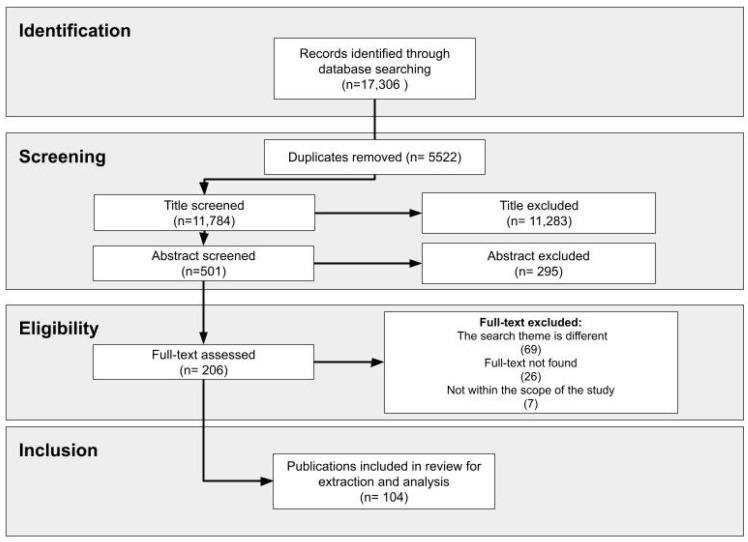
Study selection flow diagram.

**Figure 2 ijerph-19-16734-f002:**
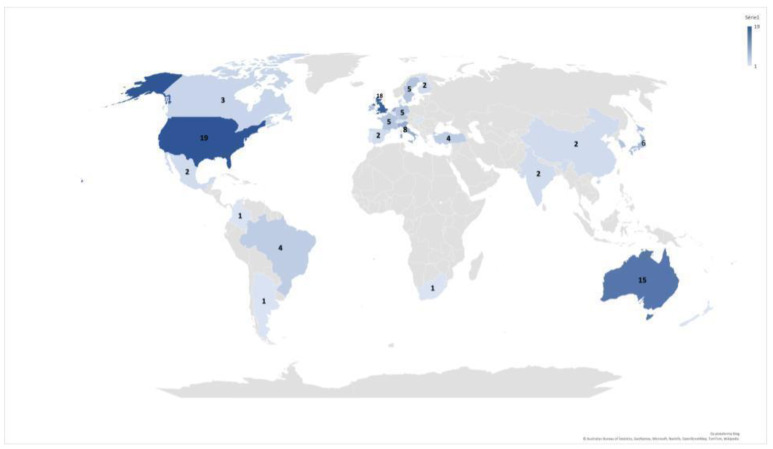
Publications on the topic around the world.

**Figure 3 ijerph-19-16734-f003:**
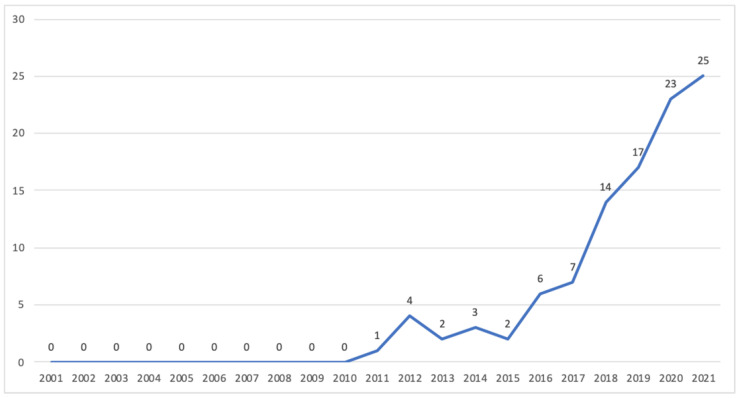
Number of studies published by year (*n* = 104).

**Figure 4 ijerph-19-16734-f004:**
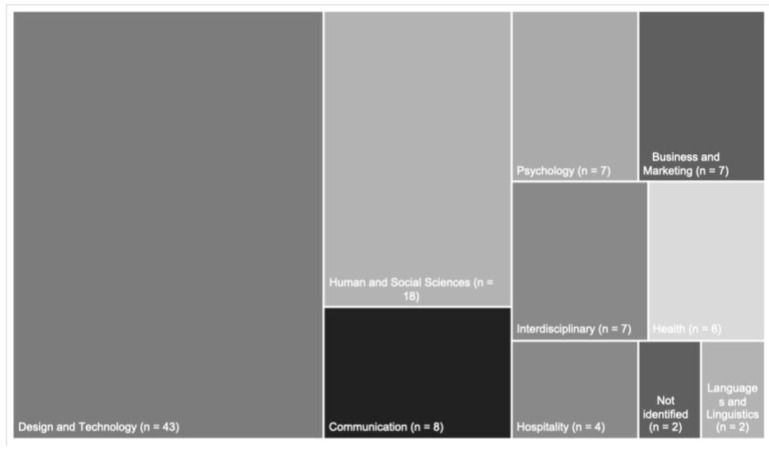
Distribution of the analyzed publications by area of knowledge.

**Figure 5 ijerph-19-16734-f005:**
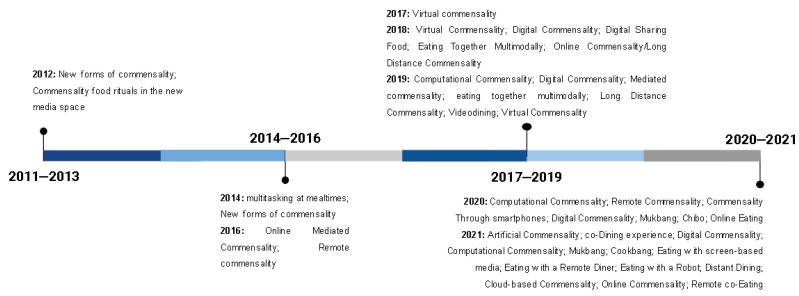
Timeline of the conceptualization of new forms of commensality.

**Table 1 ijerph-19-16734-t001:** Terms used in publications to characterize new forms of commensality.

Concepts	Description	Terms Used In The Sample Publications	Corresponding References
Commensality	In this case, the commensality addressed in the selected studies, despite being mediated by technology, is not used with a specific term that differentiates it from traditional commensality.	Commensality Cooperative eating Dinnertime ritual Eating experience Eating together Family mealtimes Food sharing Shared meals Sharing food experience Social eating Social eating experience	[10,17,30,31,32,33,34,35,36,37,38,39,40,41,42,43,44,45,46,47,48,49,50,51,52,53,54,55,56,57,58,59,60,61,62,63,64]
Remote Commensality	The practice of eating together is remote, that is, each diner is located in a different physical environment and, therefore, they meet virtually to share a meal through videoconferencing.	Cloud-based commensality Digital commensality Distant Dining Food rituals in the new media space Online Commensality/Remote co-eating Remote commensality Remote dinner or commensality across distance or long-distance commensality Virtual commensality	[49,65,66,67,68,69,70,71,72,73,74,75]
Computational Commensality	Commensality is achieved through technological artifacts used by diners or even through interaction with interactive robots.	Artificial commensality Computational commensality Digital and computational forms of commensality Digital commensality Eating together multimodally Long distance commensality Multitasking at mealtime Remote commensality	[18,29,76,77,78,79,80,81,82,83]
Digital commensality	Commensality occurs asynchronously, mediated by text, image, or video posts and instant messages	Commensality through smartphones Contemporary commensality Digital commensality (also food porn) Computational commensality Digital food sharing Mediated commensality and online mealtime socialization New forms of commensality Online mediated commensality Remote co-dining experience Virtual commensality	[16,19,28,68,84,85,86,87,88,89,90,91,92]
Commensality mediated by live broadcasts	Commensality occurs through a person having a meal (or cooking) alone and broadcasting that act to others through video platforms	Cookbang and mukbang Digital commensality Eating broadcast viewing experience Eating together multimodally Mukbang and Chibo (online meal broadcasts) Online eating Virtual commensality	[12,20,54,67,93,94,95,96,97,98,99,100,101,102,103,104,105,106,107,108,109,110,111,112,113]

**Table 2 ijerph-19-16734-t002:** Types of technologies cited in the selected publications.

Type of Technology Cited in the Publication (*n*)	Corresponding References
Social networks, applications, blogs, and websites (*n* = 32)	[10,13,17,20,32,28,36,39,40,41,46,52,54,58,62,70,84,86,87,89,90,91,92,103,104,108,114,115,116,117,118]
Video and live streaming platforms (*n* = 23)	[12,47,48,53,54,55,56,82,93,94,95,96,97,100,102,105,106,107,109,110,111,112,113]
Tabletop use prototypes (Ex: Augmented Reality) (*n* = 19)	[12,16,29,30,31,35,36,43,44,45,49,51,52,61,65,77,80,88,119]
Mobile devices (*n* = 17)	[33,34,35,38,42,56,57,59,63,73,76,73,76,79,81,99,120]
Videoconferencing tools (*n* = 14)	[15,18,24,27,50,62,65,67,68,69,81,89,101,121]
Television sets (*n* = 6)	[19,33,38,79,102,108]
Robots (*n* = 6)	[58,60,77,78,83,122]
Commercial platforms (*n* = 5)	[64,75,85,89,98]
Computers (*n* = 3)	[19,38,99]

**Table 3 ijerph-19-16734-t003:** Categories identified in studies on emerging health topics related to digital forms of commensality.

Categories (*n*)	Corresponding References
Impact on eating habits (*n* = 20)	[12,19,36,41,42,44,51,66,73,74,78,80,82,87,101,102,104,110,112,113]
Impact on mental health (*n* = 6)	[41,77,82,88,106,122]
Family and community engagement (*n* = 5)	[15,19,32,89,91]
Appreciation of local food culture (*n* = 3)	[64,68,116]
Formulation of public health policies (*n* = 2)	[72,115]
Culinary skill development (*n* = 1)	[48]
(Un)sustainability practices (*n* = 1)	[108]

## Data Availability

The data presented in this study are available upon request from the corresponding author.

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
