# Peer review of "Digital Forms of Commensality in the 21st Century: A Scoping Review"

_ijerph, 2022, doi:10.3390/ijerph192416734_

Round 1

Reviewer 1 Report

Dear Authors, overall the idea is very intriguing and up-to-date as technologies are intervening in all subjects of our lives.  There are few comments that might make your paper more clear and solid:

1. Obviously, when you started this study you had some idea of what you mean by technologies and comensality but from background section it is not quite clear. It would be great to add some clarification here.

2.I was wondering, to what extent preparing food and streeming this process could really be interpreted as "eating together"? Maybe it would be better to split form of real eating together using technologies from something slightly related to food.

3. From my perspective, the paper need clearer objectives and stronger associations between "digital" comensality and health, because finally, that would be the main effect of all studies in this field.

Author Response

Dear Reviewer,

Thank you for the suggestions and appointments. We are sure that all of them were very important and relevant for our work, whether in theoretical and technical appraisals. We tried to cover as many as of your recommendations and we hope that our answers are going to solve all the questions that were pointed out.

Dear Authors, overall the idea is very intriguing and up-to-date as technologies are intervening in all subjects of our lives.  There are few comments that might make your paper more clear and solid:

Point 1. Obviously, when you started this study you had some idea of what you mean by technologies and comensality but from background section it is not quite clear. It would be great to add some clarification here.

Response 1: We are grateful that you find our idea intriguing, we hope that our work will contribute to reflections on social relationships around food in this digital context. We also agreed to clarify what we mean about technologies and commensality and added this to the Introduction section as suggested (Lines 174-180).

Point 2. I was wondering, to what extent preparing food and streeming this process could really be interpreted as "eating together"? Maybe it would be better to split form of real eating together using technologies from something slightly related to food.

Response 2: That is a good reflection. In case you were considering practices like Cookbang, in which someone prepares food in a livestream, it is important to notice that, usually, this kind of content is followed by the consumption of the food that was prepared (Yun, Yang and Lee, 2020). With that in mind, most of these practices are considered as forms of commensality in the academic literature. The other cooking practices that were presented in the study were anchored on sociology theoretical statements (e.g Fishler, 2011) that defend that commensality is not only defined by the social interactions along the eating together time, but also extends itself for the moments before and after the meal and that goes around food, like cooking together.

Point 3. From my perspective, the paper need clearer objectives and stronger associations between "digital" comensality and health, because finally, that would be the main effect of all studies in this field.

Response 3: Perhaps the necessity of clarifying  the objectives and the associations between digital forms of commensality and health occurs since there is not — or there is with very considerable limitations — studies that strongly contribute to that result. Also, it is important to point out that we were dedicated to investigate emergent themes around the practice of digital commensality that were associated with health, considering not only its physical dimension, but its mental and social dimensions too. As a scoping review, our study aimed to contribute with the identification of gaps and to suggest opportunities for further research (Levac, Colquhoun & O’Brien (2010) — like those in the health field. Considering your appointment, we reviewed our objectives (see lines 164-165) in order to make it clear and closer to our intentions, and we also included a brief statement in the results section that reinforces our objective (see lines 503-404).

Reviewer 2 Report

Dear Authors,

The manuscript entitled “Digital forms of commensality in the 21st Century: A scoping review” deals with an interesting topic which is important from a certain point of view. However, the authors should consider the following comments to improve it.

In line 118 you mention only observational and experimental research as quantitative “articles;” however, surveys also belong to this group. I don’t think master’s theses and especially undergraduate theses provide reliable information, so I recommend you omit them from analysis. You mention that areas were identified based on the institutions’ department to which the author was linked – but in several cases, there are more than one author from more than one institution. What was your choice in those cases? Table 1 contains repetitions, see, e.g., “Social eating experience” and “Commensality” in the “Commensality” section, “Cookbang and mukbang” or “Mukbang and cookbang” and “Digital commensality” in “Commensality mediated by live broadcasts” section. In line 278 you refer to the absolute number of occurrences in Table 2, but it is missing. Please, list the types of technologies in Tables 2 and 3 in descending order based on the number of occurrences. The headers of Tables 2 and 3 are not the same (“Corresponding References” vs. “Corresponding numbers”). There are overlaps between the subsections of section 3.5, sometimes it is not clear why a topic is discussed in one subsection and not in another one. The first paragraph of the Discussion section is a command on how to write that section. The structure and clarity of that section is weak; it should be made more readable, more structured.

The wording of the paper is strange; several sentences need to be reviewed, e.g., lines 123-126, 135-136., around line 260, lines 379-380, 505-506, 516-519, and 532. The use of the term “production” instead of “results” or “studies” is especially disturbing, see, e.g., in lines 125, 443, 568, 573, 583-584, 586. There are some grammar mistakes and typos, see, e.g., lines 46, 130, 167, 208, 351, 362, 406, 459-490, 492, 505, 521, 529, and 539.  Therefore, it is highly recommended to have the manuscript proofread by a native English speaker.

Author Response

The manuscript entitled “Digital forms of commensality in the 21st Century: A scoping review” deals with an interesting topic which is important from a certain point of view. However, the authors should consider the following comments to improve it.

Dear Reviewer,

We are grateful that you find our paper interesting, we hope that our work will contribute to reflections on social relationships around food in a digital context. Thank you for your all suggestions to improve it. We are sure that all of them were very important and relevant for our work, whether in theoretical and technical appraisals. We tried to cover as many as of your recommendations and we hope that our answers are going to solve all the questions that were pointed out.

Point 1. In line 118 you mention only observational and experimental research as quantitative “articles;” however, surveys also belong to this group. I don’t think master’s theses and especially undergraduate theses provide reliable information, so I recommend you omit them from analysis. 

Response 1: We did not present these results in a good way. Thank you for pointing that out. We considered all types of qualitative studies, including surveys. Since we did not detail the types of qualitative studies, we opted to omit the “observational and experimental research” categorization that was previously mentioned. We agree that undergraduate theses do not provide reliable information and we omitted this kind of study of our analysis.

But, since master dissertation and doctoral thesis integrate the gray literature, we would like to maintain them in our study, since they can contribute with reflections, new ideas and create tendencies in the field, just like the conference papers that were included in the sample. We evidenced that the gray literature served a relevant amount of studies for this review. As a scoping review, we believe that the objective of this study was to evaluate the extension of the literature and not necessarily the quality of its studies.

Point 2. You mention that areas were identified based on the institutions’ department to which the author was linked – but in several cases, there are more than one author from more than one institution. What was your choice in those cases? 

Response 2: We are very glad with your question. Based on it, we changed how we presented these data. The studies with authors from different universities and areas are now categorized as interdisciplinary. In many cases, the authors worked in the same area, besides being from different universities and no modifications were needed.

Point 3. Table 1 contains repetitions, see, e.g., “Social eating experience” and “Commensality” in the “Commensality” section, “Cookbang and mukbang” or “Mukbang and cookbang” and “Digital commensality” in “Commensality mediated by live broadcasts” section. 

Response 3: Thank you for informing us of this mistake. We realized all the modifications in order to take out the repetitions and we also organized it in alphabetical order to facilitate the reading.

Point 4. In line 278 you refer to the absolute number of occurrences in Table 2, but it is missing. Please, list the types of technologies in Tables 2 and 3 in descending order based on the number of occurrences. The headers of Tables 2 and 3 are not the same (“Corresponding References” vs. “Corresponding numbers”). 

Response 4:We are grateful for these observations. The absolute number of occurrences was added to table 2. As requested, the tables were reorganized in descending order based on the number of occurrences and the headers were padronized.

Point 5. There are overlaps between the subsections of section 3.5, sometimes it is not clear why a topic is discussed in one subsection and not in another one. 

Response 5: Thank you for this feedback. For clarity, we have included new sentences in lines 536-551 to justify each subsection.

Point 6. The first paragraph of the Discussion section is a command on how to write that section. The structure and clarity of that section is weak; it should be made more readable, more structured.

Response 6: Thank you for alerting us to the flaw in the first paragraph. We removed the command on how to write that section. Regarding the structure and clarity of the Discussion Section, we added subtopics as a strategy to better organize the ideas presented.

Point 7. The wording of the paper is strange; several sentences need to be reviewed, e.g., lines 123-126, 135-136., around line 260, lines 379-380, 505-506, 516-519, and 532. The use of the term “production” instead of “results” or “studies” is especially disturbing, see, e.g., in lines 125, 443, 568, 573, 583-584, 586. There are some grammar mistakes and typos, see, e.g., lines 46, 130, 167, 208, 351, 362, 406, 459-490, 492, 505, 521, 529, and 539.  Therefore, it is highly recommended to have the manuscript proofread by a native English speaker.

Response 7: Also thank you for mentioning sentences that need to be reviewed and suggesting an English language revision. We updated the revised manuscript proofread by a native English speaker. We hope that our paper is more readable now.

Reviewer 3 Report

This article has only one drawback. Some references have abbreviated names of journals, but some are not. Please revise this drawback.

Author Response

Point 1. This article has only one drawback. Some references have abbreviated names of journals, but some are not. Please revise this drawback.

Response 1: Thank you for your suggestions. We reviewed all references in order to standardize them according to the journal's orientations and as requested.

Reviewer 4 Report

1.        In the section of introduction, I suggest the author should provide more information about the positive or negative effects of the development of traditional or digital forms of commensality. If could, the author could compare these forms and configure the new or innovative considerations on this issue.

2.        In the section of the scoping review approach, I suggest the author should provide practical strategies employed the methodological steps proposed by the Arksey and O’Malley model.

3.        In the section of results, I suggest the author should follow the proposed model to deal with the data and revise figure 1.

4.        This study focused on the development of digital forms of commensality. The author provided more useful information and academic discussions on this issue. But I cannot find some useful discourse on this issue and base the identified theory to analyze these results or developments. I suggest the author should focus on the related theory about the digital forms of commensality to explore these data and provide more useful theoretical and practical reflections.

Author Response

Dear Reviewer,

Thank you for the suggestions and appointments. We are sure that all of them were very important and relevant for our work, whether in theoretical and technical appraisals. We tried to cover as many as of your recommendations and we hope that our answers are going to solve all the questions that were pointed out.

Point 1. In the section of introduction, I suggest the author should provide more information about the positive or negative effects of the development of traditional or digital forms of commensality. If could, the author could compare these forms and configure the new or innovative considerations on this issue.

Response 1: Thank you for your suggestions! We added in lines 110-118 some positive and negative effects of the development of digital forms of commensality.

Point 2.  In the section of the scoping review approach, I suggest the author should provide practical strategies employed the methodological steps proposed by the Arksey and O’Malley model.

Response 2:  We are glad with your suggestion. All the strategies adopted can be identified along the PRISMA-ScR checklist. To conduct the Scoping Review, the methodological steps proposed by Arksey and O'Malley, improved by Levac and guided by the recommendations of The Joanna Briggs Institute, as mentioned in our methodology section, were used. However, to report the scoping review, we decided to follow all the steps of the PRISMA-ScR as it is a more up-to-date recommendation and is based on the suggested model. We have included a clearer mention of this in lines 202-203.

Base Reference - PRISMA Extension for Scoping Reviews (PRISMA-ScR): Checklist and Explanation | Annals of Internal Medicine (acpjournals.org)

Point 3.  In the section of results, I suggest the author should follow the proposed model to deal with the data and revise figure 1.

Response 3: As mentioned in the answer to the previous item, we decided to use the current PRISMA-ScR model. To check if the model was followed, just access the supplementary material with the checklist.

Point 4. This study focused on the development of digital forms of commensality. The author provided more useful information and academic discussions on this issue. But I cannot find some useful discourse on this issue and base the identified theory to analyze these results or developments. I suggest the author should focus on the related theory about the digital forms of commensality to explore these data and provide more useful theoretical and practical reflections.

Response 4: There are many possibilities of bringing theory to support our arguments. Our choice was based mostly in the social meanings attributed to the understanding of commensality and the role of digital technologies. That is why we gave light to our analysis by the contributions of authors such as Escobar, McLuhan, Han and others.

Round 2

Reviewer 2 Report

Dear Authors,

The manuscript entitled “Digital forms of commensality in the 21st Century: A scoping review” has been developed significantly compared to its previous version. Most of my previous concerns have been addressed; however, the authors should consider the following comments to further improve the paper.

You state that a “total of 17,306 publications were found […]. After removing 5,522 duplicates, 501 titles and abstracts were screened, and 206 documents were fully evaluated.” However, it is not clear, what was the screening criteria between the steps “removing 5,522 duplicates” and “501 titles and abstracts were screened” (i.e., how did 17,306 – 5,522 = 11,787 become 501?).

The problem of area identification based on the institutions’ department to which the author was linked is still valid – in several cases, there are more than one author from more than one institution, and your method used to handle this problem should be discussed in the manuscript as well, similarly to your answer.

You state that “Being connected through digital media does not bring people together but makes them singular” but it is not necessarily true. Please, refer to a previous study to support this statement (but you will find evidence to the opposite as well).

When you use verbatim quotes, you should include the page number as well (see lines 236-237).

Although the English language of the manuscript is much better than it was previously, there are still some typos and grammar mistakes, primarily in the newly added sentences and paragraphs; see e.g., lines 44, 73-74, 77, 78, 88-89, 106, 147, 343-345, 351, 353, 474, 573; moreover, the strange wording is still present in line 144 (“production”). Please, revise the sentence in lines 486-487.

Finally, there are some formatting issues, e.g., the font size of Table 1 should be smaller, Table 3 has too large line spaces, there is no line space between lines 556 and 557, and the citation format is not appropriate in lines 510, 566, 591, 593.

Author Response

Dear Reviewer,

We are glad that our adjustments fit some of your previous appointments. As before, we tried to do our best to cover all the new suggestions, since they were all pertinent. Please, check below our complete responses to your review.

Point 1. You state that a “total of 17,306 publications were found […]. After removing 5,522 duplicates, 501 titles and abstracts were screened, and 206 documents were fully evaluated.” However, it is not clear, what was the screening criteria between the steps “removing 5,522 duplicates” and “501 titles and abstracts were screened” (i.e., how did 17,306 – 5,522 = 11,787 become 501?).

Response 1: The screening process was made correctly, but it was not well explained in the manuscript, due to a writing problem. In fact, after removing 5522 duplicates, 11784 titles were screened and only after that 501 abstracts were screened. We made the necessary adjustments to the manuscript (line 184) and to figure 1.

Point 2. The problem of area identification based on the institutions’ department to which the author was linked is still valid – in several cases, there are more than one author from more than one institution, and your method used to handle this problem should be discussed in the manuscript as well, similarly to your answer.

Response 2: We are glad for your appointment. We included information about how we solve this problem on the manuscript (see lines 221-223).

Point 3. You state that “Being connected through digital media does not bring people together but makes them singular” but it is not necessarily true. Please, refer to a previous study to support this statement (but you will find evidence to the opposite as well).

Response 3: You brought up a very interesting point, thank you. We believe that an error occurred during the process of translating and the original meaning of the writing was lost in this case. We fixed this problem (see lines 652-66). 

In parallel, we understand your concern and we want to address this question with a few more arguments. Being digitally connected does not necessarily determine if people are together or not — and the same is true for in person reunions —, since we understand that it is not space that brings people together. Ultimately, the reflection goes in the direction of questioning and inquiring about what binds people together to be together. This can vary a lot, but we believe that issues of context operate here, of space/time relationships, of the possibilities of sharing meanings, more than the actions itselves. And these converging senses can be seen in a denser dimension: human experience with food, the table decoration, digital commensality — understood here as ordinary practices — can only connect people if they trigger devices of shared meaning. In other words, they must be interpreted by shared meanings, foreseen in the repertoire of cultures in moments of social interaction. This approach met what German hermeneutics like Gadamer have called verstehen. That is, the search for meaning that is at the heart of human experience and that gives rise to otherness and difference in the way we understand the world.

Point 4. When you use verbatim quotes, you should include the page number as well (see lines 236-237).

Response 4: Thank you for your suggestion. We included it. 

Point 5. Although the English language of the manuscript is much better than it was previously, there are still some typos and grammar mistakes, primarily in the newly added sentences and paragraphs; see e.g., lines 44, 73-74, 77, 78, 88-89, 106, 147, 343-345, 351, 353, 474, 573; moreover, the strange wording is still present in line 144 (“production”). Please, revise the sentence in lines 486-487.

Response 5: We are grateful for these observations. The manuscript was revised once again. We hope that our paper is more readable now. 

Point 6. Finally, there are some formatting issues, e.g., the font size of Table 1 should be smaller, Table 3 has too large line spaces, there is no line space between lines 556 and 557, and the citation format is not appropriate in lines 510, 566, 591, 593.

Response 6: We fixed all the formatting issues on the tables and on the citation format mentioned in your comment. We also looked for other formatting issues that might have appeared. Thank you for pointing that out.